# Pore configuration landscape of granular crystallization

M. Saadatfar[1], H. Takeuchi[2], V. Robins[1], N. Francois[3] & Y. Hiraoka[4]

Uncovering grain-scale mechanisms that underlie the disorder–order transition in assemblies of dissipative, athermal particles is a fundamental problem with technological relevance. To date, the study of granular crystallization has mainly focussed on the symmetry of crystalline patterns while their emergence and growth from irregular clusters of grains remains largely unexplored. Here crystallization of three-dimensional packings of frictional spheres is studied at the grain-scale using X-ray tomography and persistent homology. The latter produces a map of the topological configurations of grains within static partially crystallized packings. Using numerical simulations, we show that similar maps are measured dynamically during the melting of a perfect crystal. This map encodes new information on the formation process of tetrahedral and octahedral pores, the building blocks of perfect crystals. Four key formation mechanisms of these pores reproduce the main changes of the map during crystallization and provide continuous deformation pathways representative of the crystallization dynamics.

[1] Department of Applied Mathematics, Research School of Physics and Engineering, The Australian National University, Canberra, Australian Capital Territory, Australia. [2] Mathematics Department, Graduate School of Science, Tohoku University, 6-3, Aramaki Aza-Aoba, Aoba-ku, Sendai 980-8578, Japan. [3] Centre for Plasmas and Fluids, Research School of Physics and Engineering, The Australian National University, Canberra, Australian Capital Territory, Australia. [4] WPI-Advanced Institute for Materials Research (WPI-AIMR), Tohoku University. 2-1-1, Katahira, Aoba-ku, Sendai 980-8577, Japan. Correspondence and requests for materials should be addressed to M.S. (email: mohammad.saadatfar@anu.edu.au) or to H.T. (email: hiroshi.takeuchi.s6@dc.tohoku.ac.jp) or to N.F. (email: nicolas.francois@anu.edu.au).

Ordering and crystallization in a thermal system can be achieved by cooling it down; in contrast, in macroscopic sphere packings high levels of energy need to be injected to overcome the natural propensity of these materials to form amorphous structures[1–5]. Despite the significance of regular packings in a wide range of fields, from fundamental physics to granular processing[6,7], the basic mechanism of crystallization in this highly dissipative material is still unknown. Indeed the transition from disordered to ordered packings triggers a wide range of geometrical, topological and mechanical changes at multiple length scales, some of which have just recently been uncovered[4,8–13]. To describe this complexity and since sphere packings are comprised of numerous particles, it is tempting to use a statistical approach.

In analogy with equilibrium statistical mechanics, Edwards and co-workers have laid the foundations of a statistical mechanics for amorphous jammed packings[14–17]. Since macroscopic granular packings are dissipative due to friction, it has been argued that the volume of the system and a variable characterizing the mechanical state should replace the energy as key macroscopic descriptors[18,19]. Such a statistical approach hinges on the definition of the space of the possible jammed configurations of grains[16,20]. This space can be considered as the analogue of the phase space in equilibrium thermodynamics. Nevertheless drawing this analogy is no trivial matter, because in granular materials, friction introduces a sharp nonlinearity in grain contact laws, which challenges our understanding of mechanically stable structures[21–24]. As a consequence, the nature and description of the space of stable jammed states remain outstanding questions[16,19,25–29].

Even less is known about packings with larger densities. These packings are usually formed under strong vibrations and a sharp structural transition is observed at Bernal's density ($\phi_{\text{Bernal}} \approx 0.64$)[4,11,25]. The latter signals the appearance of crystalline clusters in the bead assembly. Recent advances show that the formation of a granular crystal might be described according to a thermodynamic-like process in idealistic three-dimensional (3D) packings[10]. However, experimental studies on the disorder–order transition in 2D granular layers have demonstrated both striking resemblances and profound differences with thermal systems[30–33]. To better understand and support any thermodynamic-like approach, it would be interesting to be able to measure the phase space of the grain configurations representative of the crystallization of realistic packings (that is, polydisperse and frictional). Such a measurement is an important prerequisite to the definition of a robust statistical description.

Here we experimentally study 3D packings made of cohesion-less macroscopic spheres. Recent developments in X-ray computed tomography (XCT)[4,12,13,34] and topological data analysis[35,36] are utilized to describe at the grain-scale the cavities that exist and are mechanically stable in large partially crystallized packings. Our analysis is based on persistent homology (PH), a technique that allows the characterization and quantification of geometrical shapes in structure[37–39]. In comparison with classical spatial tessellation techniques (such as Voronoi or Delaunay methods), PH offers a clearer description of the population of cavities and a more comprehensive view of crystallization-driven structural changes. More precisely, it gives us persistence diagrams, akin to a topological phase space, that represent all the grain configurations and patterns within a packing.

In this study, we explore and characterize the topological changes of partially crystallized packings at mechanical equilibrium over the density range $\phi = (0.60, 0.72)$. The shape of the diagrams highlights four key ordering mechanisms that govern the formation of regular tetrahedral and octahedral patterns of grains: the two basic components of perfect crystalline arrangements. By using numerical simulations, we show that these mechanisms are relevant to describe the dynamics of the quasi-static 'melting' of a granular crystal. In the context of our experimental results, it shows that persistence diagrams are able to capture interesting features of the crystallization dynamics despite being computed on static mechanically stable states. We discuss some of the consequences of these findings in terms of the ergodicity of this out-of-equilibrium transition.

## Results

**Experimental procedure**. Six experimental granular packings are produced each containing monosized acrylic beads (diameter $D = 1$ or $1.62$ mm, polydispersity $= 0.025$ mm, polydispersity is defined as the standard deviation in grains' effective diameters, see details in Methods section). Table 1 provides experimental details on each packings analysed. In particular, we have analysed: (i) a fully disordered packing with a packing density of $\phi = 0.635$, produced by pouring beads into a cylindrical container and (ii) partially ordered packings produced in cylindrical and spherical containers using a vibrational protocol (see Fig. 1). Interestingly a cylindrical packing and a spherical one have the same packing density of $\phi = 0.685$.

Details on the experimental procedures used to produce amorphous and partially crystallized packings can be found in the Methods section and in refs 4,12,26. Helical XCT is utilized to image the internal 3D structure of the packings at a cubic voxel size of $\approx 30$ micrometres[4,34]. Our experiments combine XCT and 3D image analysis to accurately determine grain centroids with a precision of $\approx 10^{-3}$ µm and grain diameters with a precision of $\approx 5.10^{-2}$ µm (see Methods section and refs 12,13). In previous studies dealing with packings obtained by vibration-driven densification, an abrupt structural transition has been identified at Bernal's density $\phi_{\text{Bernal}} \approx 0.64$ (refs 4,12). This transition corresponds to the onset of crystallization in sphere packings.

Figure 1 shows 3D renderings of the internal structure of partially crystallized packings obtained in spherical and cylindrical containers. Figure 1b illustrates that both random and crystalline phases coexist in the packing; in the crystalline areas, defects are observed in the form of dislocations and grain vacancies. Our analyses have been carried out over the entire packing structures as well as 23 non-overlapping subsets each containing 4,000 beads. These subsets are from the inner region of the packings, four sphere diameters away from the container walls. The amorphous packing ($\phi < \phi_{\text{Bernal}}$) has a homogeneous structure, therefore subsets from this packing have similar densities. In contrast, the partially ordered packing shows spatial heterogeneity in its structure. Consequently, the subsets extracted from it have a wide range of density ranging from $\phi = 0.58$ to $\phi = 0.73$.

---

**Table 1 | Summary of the experimental packings analysed in this study.**

| Container | N | $\phi$ | D (mm) |
|---|---|---|---|
| 1 Cylinder | 150,000 | 0.60 | 1.00 |
| 2 Cylinder | 156,315 | 0.63 | 1.00 |
| 3 Cylinder | 61,005 | 0.66 | 1.62 |
| 4 Cylinder | 216,722 | 0.685 | 1.00 |
| 5 Spherical | 86,281 | 0.685 | 1.00 |
| 6 Spherical | 86,706 | 0.71 | 1.00 |

N is the number of grains used for the analysis; $\phi$ is the global packing density; D is the grain diameter.

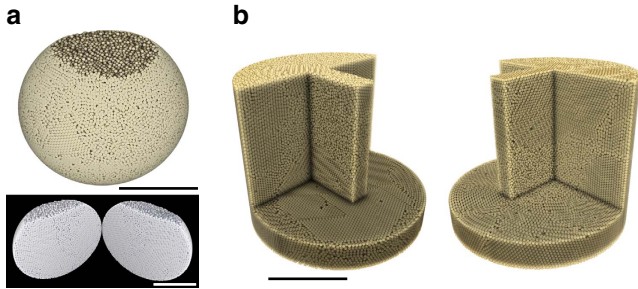

**Figure 1 | 3D rendering of spherical and cylindrical experimental packings.** Internal structure of partially crystallized packing with a global packing density of $\phi = 0.685$ in (**a**) a spherical container and (**b**) a cylindrical container. The scale bar length is $\approx 25$ mm. In (**a**), the spherical packing contains 80,000 beads. In panel (**b**), two different 3D sections of a cylindrical packing made of 200,000 beads. Panel (**b**) highlights the heterogeneous structure of our packings composed of both random and highly crystalline clusters. By comparing the two 3D sections, it also shows that the rich polycrystalline structures of our packings are characterized by defects, such as grain vacancies and dislocations.

**Homology and PH**. Homology is an algebraic method for studying topological features of geometrical objects. The starting point for computing homology is a complex, $\mathcal{C}$, essentially a collection of building blocks whose union is the shape of interest. In a simplicial complex, the building blocks are points, edges, triangles, tetrahedra and higher $k$-dimensional simplices. A $k$-chain is a formal sum of $k$-dimensional simplices. A $k$-cycle is a chain whose boundary is empty (the sum of its faces cancel out). The homology groups $H_k$ encode equivalence classes of these $k$-cycles. In more physical terms, homology groups $H_k$ describe the presence of $k$-dimensional holes in an object (see Methods section). In particular, the $H_2$ group describes the presence of 2-cycles, that is, enclosed cavities, within a structure.

In this work, we are interested in the properties of the $H_2$ group obtained from sphere packings and its varying properties as the packing density increases. To this end, the concept of PH is used. PH extends traditional homology by tracking how the homology groups change when a control parameter $\alpha$ is varied. This control parameter is known as the filtration parameter. PH has become an increasingly useful tool for studying the shape of data in a broad range of areas, such as sensor networks[40], high-dimensional data mining[35], digital images[41–43], biochemistry[44], materials science[39,45] and recently for describing granular materials[46–48]. Let us illustrate how the method works in the case of bead packings.

First, the packing data are specified by the coordinates of the centre of each bead and its measured radius $r$, extracted from micro-CT images[26]. The main idea is now to consider that the bead radius is a free variable while the bead centres remain fixed. In a sense, we obtain a virtual packing, where the virtual bead radius is now the filtration parameter $\alpha$ that can vary freely from 0 to $\infty$. In mathematical terms, the problem concerns the characterization of the union of balls of radius $\alpha$ growing around each experimentally measured bead centre $x$: $X(\alpha) = \cup \, B(x, \alpha)$. PH tracks the changes in the topology of $X(\alpha)$ that occur as the ball radius increases from zero to $\infty$. For $0 < \alpha < r$, $X(\alpha)$ is a disjoint union of balls. For $\alpha \geq r$, bead contacts are resolved and $X(\alpha)$ becomes connected, initially with many cavities, which are then filled in as $\alpha$ increases. From the physics viewpoint, it is important to note that PH characterizes primarily the presence of cavities and as such balls are considered as transparent objects that can overlap (see Fig. 2a).

Using this technique, each cavity in the packing can thus be identified by its birth value $\alpha = b$ (when four or more surrounding beads come into contact) and its death value $\alpha = d$ (when its surrounding beads become so large that the cavity is filled). It is common practice to represent the birth and death value for each cavity in a persistence diagram, named $PD_2$. $PD_2$ contains all the $(b,d)$ pairs of a given packing, each corresponding to topologically distinct cavities that exists in the interstices between the beads. $PD_2$ contains information not only about the number of cavities but also on the nature of each cavity/pore or on the frequency of occurence of patterns (regular-distorted) that have been detected within the packing. As such, $PD_2$ offers new ways to describe the phase space of stable grain configurations.

An interesting asset of $PD_2$ is that certain local cavities with a specific motif can be clearly and simultaneously identified in the same diagram. For instance, the simplest and smallest pore is that formed inside four beads closely packed as a regular tetrahedron. From a topological viewpoint, a tetrahedral pore is born when $\alpha$ (the filtration parameter) reaches the circumradius of an equilateral triangle and dies when $\alpha$ is the circumradius of the regular

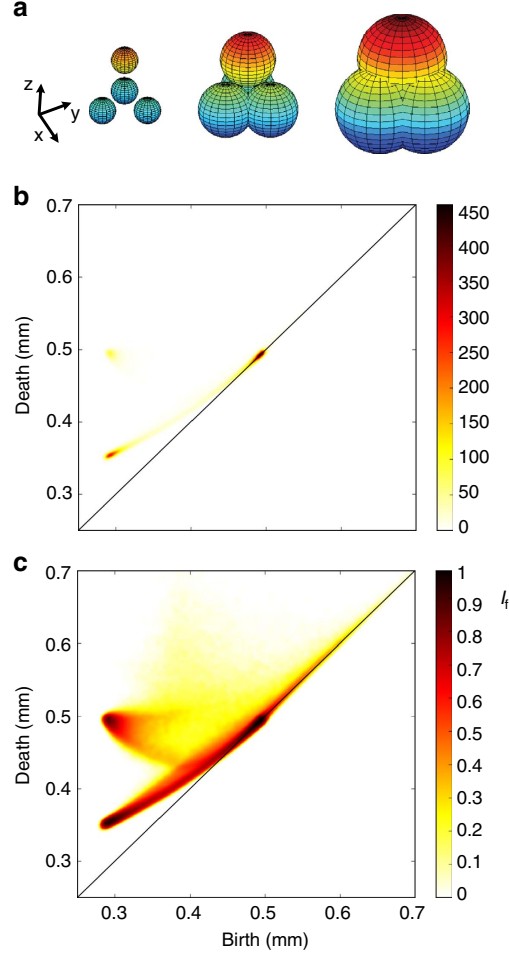

**Figure 2 | Persistence diagrams.** (**a**) Tetrahedral configuration of mathematical balls with growing radius $\alpha$. Colour changes from blue to red with increasing $z$ value. (**b**) Persistence diagram $PD_2$ as a probability density function of pores indicating via the colour map the occurence rate $P$ of a given grain configuration at a specific $(b,d)$ pair. $PD_2$ is plotted for a packing with a density of $\phi = 0.685$. (**c**) $PD_2$ for the same packing, the rate of occurence is now expressed in terms of the frequency index $l_f = \log((\log(P + 1)) + 1)/l_{fM}$ to highlight the fine details of the diagram.

tetrahedron. This mathematical definition ensures that a regular tetrahedral cavity is uniquely identified by the point $(b_t, d_t) = (2r\sqrt{\frac{1}{3}}, 2r\sqrt{\frac{3}{8}})$ in $PD_2$. Another simple pore is that formed by six beads closely packed as a regular octahedron. This type of cavity is uniquely identified by the point $(b_o, d_o) = (2r\sqrt{\frac{1}{3}}, r\sqrt{2})$ in $PD_2$ (see Methods section). On a technical note, the persistence diagrams presented here are computed using a weighted alpha filtration related to a weighted Voronoi decomposition. It uses the power distance instead of the absolute distance between the bead centres. This approach allows us to take into account the small, yet finite, polydispersity of the grains. In the $PD_2$ based on the power distance, regular tetrahedron and octahedron cavities are located at $(b_t, d_t) = (\sqrt{\frac{4}{3}r^2 - r^2}, \sqrt{\frac{3}{2}r^2 - r^2})$ and $(b_o, d_o) = (\sqrt{\frac{4}{3}r^2 - r^2}, \sqrt{2r^2 - r^2})$. In our experiments, the average bead radius is $\langle r \rangle = 0.5$ mm, therefore:

$$(b_t, d_t) = (0.288, 0.353), \quad (b_o, d_o) = (0.288, 0.5)$$

when expressed in millimetres.

The basis for computing PH is the alpha shape (subsets of the Delaunay tessellation (DT) obtained by adding triangles and tetrahedra ordered by their circumradii) and therefore it integrates the geometry as well as local topology of grain structures. In short, PH naturally incorporates the relevant local correlations in bead positions to reveal short- and medium-range order and even some global structures (such as percolating length scales[48]). Another asset of the PH approach is that the detection of cavities is a robust process that is only weakly affected by the finite resolution and precision of our experimental measurements[47,49]. In the following section, we present persistence diagrams $PD_2$ computed on our experimental granular packings over a broad range of packing densities $0.60 < \phi < 0.73$.

Figure 2 shows a typical persistence diagram $PD_2$ for a sphere packing at density $\phi \approx 0.685$. In Fig. 2b,c, $PD_2$ is plotted as a probability density function indicating via the colour map the occurence rate $P$ of a grain configuration identified by a given $(b, d)$. In Fig. 2c, the occurence rate is expressed in terms of the frequency index $I_f = \log((\log(P + 1)) + 1)/I_{fM}$ to highlight fine details of the diagram $(I_{fM} = \text{Max}(\log((\log(P + 1)) + 1)))$ for the data set). In the following, we always use this loglog scale in order to highlight fine details of $PD_2$.

## PD$_2$ in different container geometry and changes versus $\phi$.
First, we have computed $PD_2$ diagrams on partially crystallized packings produced in different container geometry (see Fig. 3). When computed on these large structures (each packing contains at least 60,000 beads), the $PD_2$ diagrams reveal a generic domain of existence whose shape seems independent of the container geometry. Nevertheless, there are differences in how this domain is populated in these different packings. These changes can be understood by the different level of compaction of these packings.

Figure 4 shows representative persistence diagrams measured from sphere packings produced in cylindrical containers, with internal structures ranging from amorphous to nearly perfect crystalline arrangements. Below Bernal's limit ($\phi \approx 0.64$), a great number of cavities are confined within a curved strip, one end of which is pointing to the location of the tetrahedral coordinate $(b_t, d_t) = (0.288, 0.353)$. We note the presence of a less populated region located at higher death values ($d > r \times \sqrt{2}$) with a peak around $(b, d) = (0.4, 0.7)$ as seen in Fig. 4a,b. Each cavity in this region is irregular, larger than octahedral pores and is a connected chain of many pores[48]. As crystallization takes place for $\phi > 0.64$ (Fig. 4c,d), the diagram assumes a distinctive shape

with a horn-shaped feature converging towards the octahedral coordinate at $(b_o, d_o) = (0.288, 0.5)$. Ultimately (for $\phi > 0.72$), the entire population of cavities collapses onto two locations (or hot spots) that correspond to the regular tetrahedral and octahedral cavities (see Fig. 4d). This reflects the fact that perfectly crystalline sphere packings consist uniquely of a periodic arrangement of tetrahedral and octahedral cavities. A perfect crystalline packing configuration has a packing density of $\phi = 0.74$.

The evolution of the persistence diagram versus $\phi$ shows that PH is an extremely sensitive tool to explore structural and topological changes that accompany granular crystallization. Moreover, we emphasize that $PD_2$ is a particularly relevant tool for analysing the geometry of sphere packings as it lays out an efficient way to distinguish structures with almost identical packing fraction but different topology.

## Quantification of tetrahedral and octahedral cavities.
Quasi-regular tetrahedral or octahedral grain configurations are central to our understanding of packing structure[25,50,51]. They play a key role in the description of geometrical frustration in amorphous packings $\phi < 0.64$ or of the growth of crystalline clusters at higher packing density[4,52].

PH offers a new way to quantitatively measure the number of tetrahedral and octahedral cavities in sphere packings. Indeed, by using $PD_2$, it is possible to separate tetrahedral and octahedral cavities according to their proximity to the tetrahedral and octahedral hot spots. Not only due to the sphere polydispersity but also due to the presence of defects (such as grain boundaries and dislocations) in large crystalline clusters (see Fig. 1b), the tetrahedral and octahedral cavities formed during crystallization remain slightly irregular even at the highest packing density and therefore do not collapse perfectly on the two points $(b_t, d_t)$ and $(b_o, d_o)$. To count the number of quasi-regular cavities, rectangular regions are defined around the hot spots (see Fig. 5a) such that they contain 50% of the total $PD_2$ points in the most crystalline yet still imperfect packing at $\phi = 0.73$.

Figure 5b shows the evolution of the number $N_{tetra}$ and $N_{octa}$ of tetrahedral and octahedral cavities (normalized by the total number of cavities) as a function of packing density $\phi$ measured by counting the number of $(b, d)$ points inside these rectangular regions. In loose random packings ($\phi = 0.59$), we measure $N_{tetra} \approx 3\%$. $N_{tetra}$ increases with $\phi$ and its rate of increase doubles beyond $\phi = 0.64$. Below Bernal's packing density, there are almost no octahedral cavities; however, as the packing density increases beyond $\phi = 0.64$, their number rises sharply.

Figure 5c shows the proportions $P_{tetra}$ and $P_{octa}$ of tetrahedral and octahedral pores among the cavities identified as quasi-regular. The solid horizontal lines are drawn at the values of tetrahedral and octahedral cavities for a perfect crystalline structure (that is, $P_{tetra} \approx 0.67$ and $P_{octa} \approx 0.33$). Below $\phi \approx 0.64$, quasi-regular cavities are almost entirely tetrahedral cavities. A clear transition can be observed at $\phi \approx 0.64$ but Fig. 5c also highlights another sharp transition occuring at $\phi \approx 0.675$. Beyond $\phi \approx 0.72$, $P_{tetra}$ and $P_{octa}$ reach a plateau at the predicted crystalline values.

The transition at $\phi \approx 0.64$ signals the crystallization onset. The two other transitions occurring at $\phi \approx 0.68$ and $\phi \approx 0.72$ have recently been reported and studied in the context of geometrically frustrated patterns in partially crystallized packings[4]. For instance, dense rings made of five quasi-regular tetrahedra, a ubiquitous feature of amorphous packings, disappear completely at $\phi \approx 0.68$. These transitions also play an important role in the mechanical stability of these packings[10,12,13].

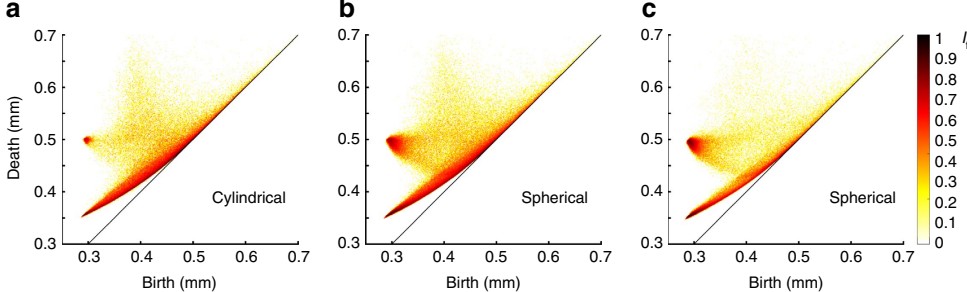

**Figure 3 | Persistence diagram of experimental packings.** Persistence diagrams $PD_2$ of partially crystallized packings produced in: (**a**) a cylindrical container with a density $\phi = 0.66$ ($N = 61{,}000$ and $D = 1.62$ mm, note that the ($b,d$) coordinates have been rescaled to be directly comparable with data shown in panels (**b**,**c**) for which $D = 1$ mm), (**b**) a spherical container with a density $\phi = 0.685$ ($N = 86{,}000$ and $D = 1$ mm), (**c**) a spherical container with a density $\phi = 0.71$ ($N = 86{,}000$ and $D = 1$ mm).

We note that the evolution of $N_{\text{tetra}}$ is different to that reported in analysis based on maximal edge distortion[11]. It shows the novelty of this topological definition of quasi-regular cavity which still captures essential structural transitions during packing crystallization[4].

## Discussion

The persistence diagram can unambiguously identify the presence of tetrahedral and octahedral cavities in packings. The evolution of $PD_2$ seen in Fig. 4, however, shows that there are populated regions in both disordered and partially ordered packings that do not correspond to regular cavities. To shed light on the origin of this configuration landscape, we now consider basic formation/deformation mechanisms of tetrahedral and octahedral cavities and investigate their signatures in $PD_2$ as the packing density changes. Importantly, the evolution of ($b$, $d$) coordinates of these four cavity deformations can be derived analytically using a combination of dihedral angles and grain radius (see Methods section).

We begin by considering two types of tetrahedral deformation:

(i) the first deformation, D1, consists of letting one bead of a tetrahedron roll across the saddle area formed by two neighbours, while the other beads stay fixed. Topographic top and side views of the deformation D1 are shown in Fig. 6b.

(ii) the second deformation, D2, is a symmetric process that transforms a regular tetrahedron into a flat square configuration. Topographic top and side views of this deformation are shown in Fig. 6b.

Animated visualizations of D1 and D2 deformations are provided in Supplementary Movies 1 and 2.

Now we consider two types of octahedral deformations:

(i) the first deformation, D3, consists in lengthening two opposite edges of a regular octahedron (symmetrically), while keeping all other edges equal to $2r$, until two edge-adjacent tetrahedra are formed as shown in Fig. 6c.

(ii) the second deformation, D4, is reminiscent of the first tetrahedral deformation D1: one of the beads rolls along the saddle area formed by neighbouring beads (see Fig. 6c).

Animated visualizations of D3 and D4 deformations are provided in Supplementary Movies 3 and 4.

The analytically computed birth–death curves of the above four deformation scenarios are superimposed on $PD_2$ of a partially crystallized packing with density of $\phi = 0.685$ (Fig. 6a). The

locations of these curves match well with the boundaries of the domain of existence measured via $PD_2$. This is a significant result because of the following two reasons:

(i) our topological diagrams actually characterize different mechanical equilibrium states of static packings.

(ii) in theoretical approach, it is commonly assumed that the configuration space of jammed matter is discrete to ensure the actual jamming of a structure[16].

Therefore, it is quite remarkable that continuous deformation mechanisms seem to accurately reproduce important features of the configuration space $PD_2$ over a broad density range of $0.64 < \phi < 0.73$. It suggests that $PD_2$ captures some representative features of the crystallization dynamics despite being computed on static states at mechanical equilibrium.

To test the relevance of this hypothesis, we have performed numerical simulations in which perfectly crystalline layers of frictional grains are sheared until the packing becomes disordered. It allows us to dynamically track the 'melting' of a granular crystal.

Initially, a packing of 6,000 mono-disperse spheres is formed according to a face-centre-cubic (FCC) motif. The bottom layer of grains is then set into motion at a constant shear rate $\dot{\gamma}$ (see Fig. 7b). The forces (both normal and frictional) between the grains and grain displacement are computed at each time step by a discrete element method that uses Hertz–Mindlin contact model[12,13]. During the crystal 'melting', a characteristic shear rate is evaluated as $\dot{\gamma} = v_{\text{g}}/(4d)$, where $v_{\text{g}}$ is the velocity of shearing boundary. The simulated process is quasi-static in the sense that the characteristic time of decrease of $\phi$ is more than 10 times larger than the characteristic time $1/\dot{\gamma}$ associated with the shear deformation applied (see Fig. 7a). During the melting, the packing remains dense with a packing density that stays larger than $\phi = 0.54$ during the transition. Moreover, a high number of mechanical contacts is maintained throughout the process: the average mechanical coordination number $Z$ is initially 12 and remains $> 4$ during the evolution. Figure 7c–f show the temporal evolution of the $PD_2$ as this virtual packing becomes more and more disordered. The signature of this quasi-static transition in the topological space is highly similar to the one measured in partially crystallized packings at mechanical equilibrium.

These results reveal a connection between our experimental results and a quasi-static 'melting'; this could be interpreted as the existence of a form of ergodicity in the disorder–order transition of granular packings. On this basis, experimentally measured $PD_2$ diagrams could actually reveal structural deformation paths that are statistically representative of the crystallization dynamics. On a related note, recent experimental studies on the same packings

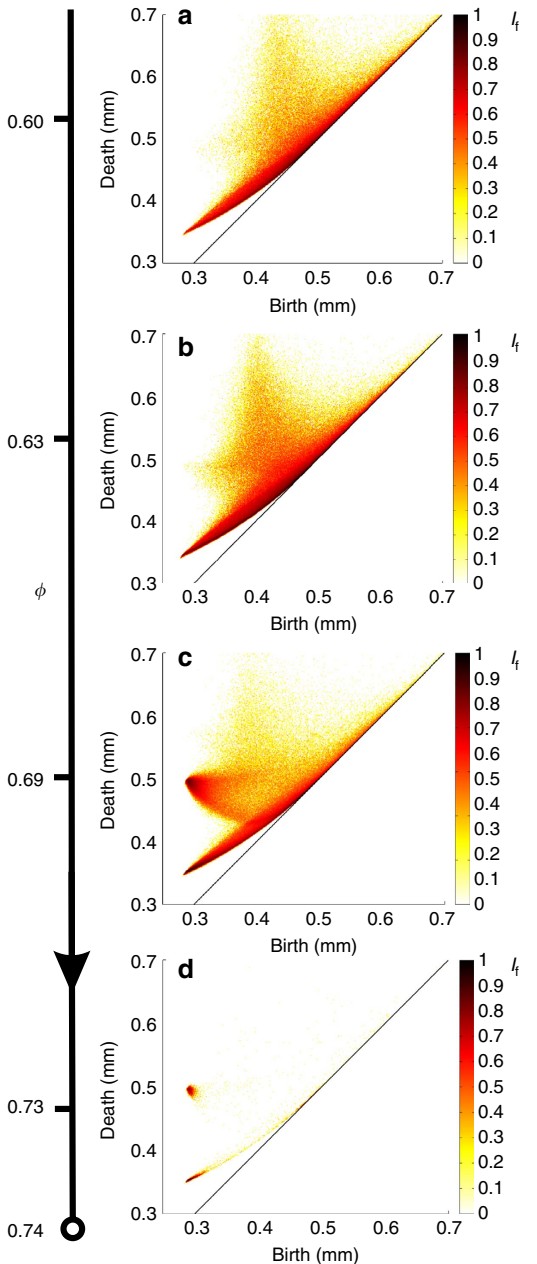

**Figure 4 | PD₂ of disordered, partially ordered and highly ordered sphere packings.** Representative $PD_2$ of sphere packings with density ranging from $\phi = 0.60$ to $\phi = 0.73$. Crystallization onset occurs at $\phi \approx 0.64$. The diagrams in panels (**a,b**) have been computed over more than 150,000 beads and 500,000 cavities. The diagram in (**c**) has been computed over more than 200,000 beads and 800,000 cavities. The diagram in (**d**) has been computed over more than 4,000 beads and 20,000 cavities.

by some of the authors have reported some analogies between the crystallization-driven changes in the packing mechanical structure and a first-order transition[12,13]. Finally, we have tested other pore deformation mechanisms that allow more contacts to be modified, but none of them describe accurately the evolution of $PD_2$; for most of them, the birth–death curve was outside the domain of existence.

We now describe how the deformation curves defined above represent boundaries in the local pore configuration landscape. The D1 and D2 curves run along the lower and upper boundaries of a region comprised of many cavities, and they meet at the

location $(b_t, d_t) = (0.288, 0.353)$ of the regular tetrahedral cavity (Fig. 6a). D3 and D4 curves meet at the octahedral hot spot $(b_o, d_o) = (0.288, 0.5)$ and they provide the lower and upper boundaries to a horn-shaped region that appears for packing density $\phi > 0.64$ (Fig. 4). We note that the D3 curve is made of two branches, one of which is similar to the D1 curve and ends at the tetrahedral hot spot. We emphasize that the highlighted boundaries correspond to the most populated curves that reflects their importance in the densification process via crystallization. By combining these new topological data and recent discoveries in the geometry of packings[4,11], a grain-scale picture of the disorder–order transition emerges over the whole density range explored.

The curves D1 and D2 clearly delineate a highly populated strip, which we identify as a region made of weakly distorted tetrahedral cavities. The extent of this domain confirms the variety of shapes and ubiquity of these distorted tetrahedra in amorphous packings $\phi < 0.64$ (see Fig. 4a). Although this strip becomes less and less populated in favour of the tetrahedral hot spot as crystallization proceeds $\phi > 0.64$, we note that it persists up to the density $\phi \approx 0.7$ (see Figs 4 and 8). It has recently been shown that the formation of polytetrahedral aggregates composed of weakly distorted tetrahedra is a geometrical principle of densification for amorphous frictional packings and a resilient feature in partially crystallized structures up to $\phi = 0.72$ (refs 4,11,51).

Within the density range $0.64 < \phi < 0.73$, the number of cavities that occupy the horn-shaped region delimited by the curves D3 and D4 increases sharply (see Figs 4 and 8 and Methods section). The two scenarios D3 and D4 suggest that two structural mechanisms are at play: (i) two edge-connected tetrahedra can coalesce to form an octahedral cavity, (ii) irregular cavities transform into initially distorted octahedra and with further increase of $\phi$, they form regular octahedra as grains are forced to compact closely. The D3 scenario reveals that edge-connected tetrahedra play a role in the crystal formation. This role was not considered in recent characterization of packings based on geometrically frustrated patterns, that is, patterns that are made of face-adjacent tetrahedra[4,11,51]. It is important to note that the distinction between face-adjacent and edge-connected tetrahedra is instrumental in detecting the presence of either hexagonal closely packed or FCC crystalline patterns (hexagonal closely packed and FCC patterns are, respectively, built upon face-adjacent or edge-connected tetrahedra). To date, the selection of one pattern or the other remains an outstanding question for crystal formation in athermal systems[8,53]. Recent studies on macroscopic packings have revealed that both crystalline motifs coexist in highly crystalline packings with a slight preference for the FCC pattern[3,13]. In our polycrystalline packings, the average ratio of FCC motifs over the total number of crystalline motifs is $\approx 2/3$ beyond $\phi > 0.72$ (ref. 13). With this polycrystalline picture in mind, we note that there is something singular about the scenario D3: its extremal states (an octahedral cavity and two edge-connected tetrahedra) can exist in perfect crystalline arrangements. The scenario D3 is therefore reversible and it suggests that edge-adjacent tetrahedra can be formed via the distortion of octahedral cavities. This is a tantalizing grain-scale mechanism to understand the prevalence of FCC crystal at high packing density in our packings.

In conclusion, we have shown that important features of the configurational landscape of 3D granular packings can be characterized using topological diagrams from PH. These diagrams change dramatically during the crystallization and allow to describe the formation of tetrahedral and octahedral cavities. Three structural transitions have been detected and related to those recently reported in the context of geometrically frustrated

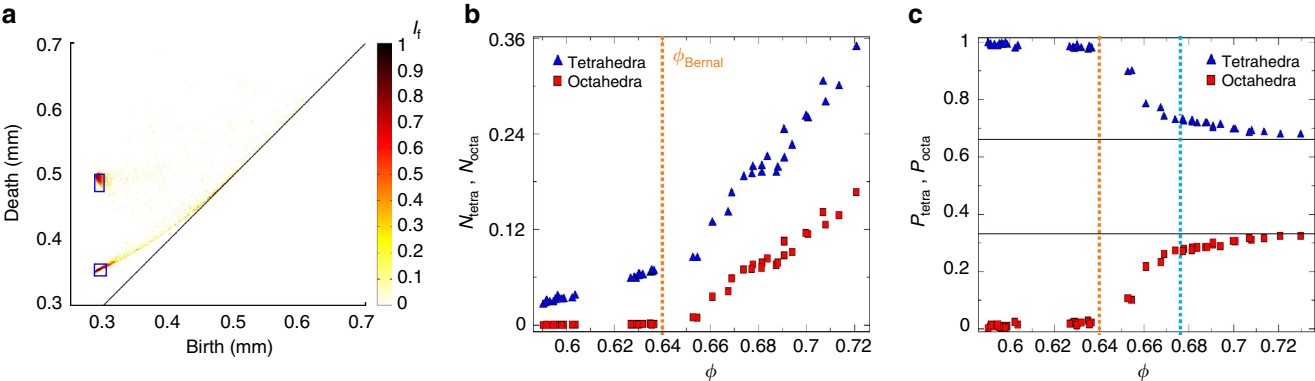

**Figure 5 | Quantification of regular cavities versus $\phi$ based on PD$_2$.** (a) PD$_2$ for a sphere packing with a density of $\phi = 0.73$. The blue squares indicate the regions of interest used to count the number of quasi regular tetrahedral and octahedral cavities. The size of these rectangular regions are defined by birth and death associated with grain polydispersity ($D \pm 0.025$ mm) around (birth, death) tetrahedral and octahedral cavities. (b) Numbers $N_{tetra}$ and $N_{octa}$ of quasi regular tetrahedral and octahedral cavities (normalized by the total number of cavities) versus packing density $\phi$. (c) Proportion $P_{tetra} = N_{tetra}/(N_{tetra} + N_{octa})$ and $P_{octa} = N_{octa}/(N_{tetra} + N_{octa})$ of quasi regular tetrahedral and octahedral cavities versus packing density $\phi$.

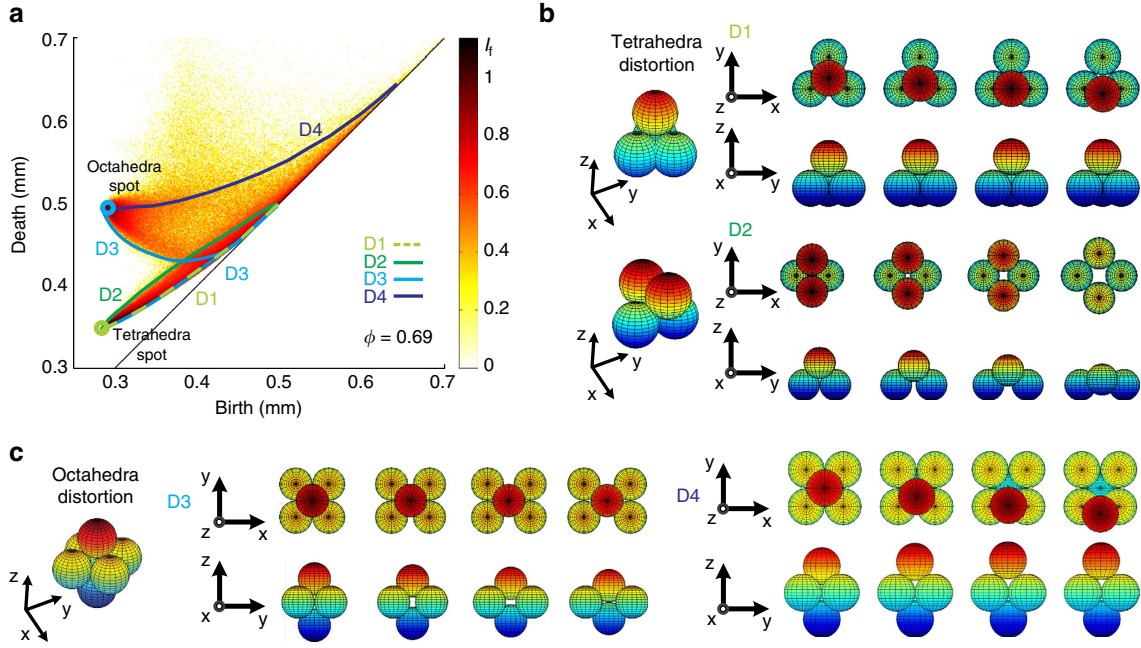

**Figure 6 | Grain-scale tetrahedral and octahedral formation/deformation scenarios.** (a) PD$_2$ of a partially crystallized sphere packing with density $\phi = 0.685$. The superimposed curves correspond to analytically computed birth–death curves of the deformation scenarios shown in panels (**b,c**). (**b**) Top and side views of D1 and D2 deformations scenarios of a tetrahedral cavity. (**c**) Top and side views of D3 and D4 deformation scenarios of an octahedral cavity. The colour code indicates the relative height of sections of the grain with respect to the horizontal median plane.

patterns[4] and mechanical stability[10,12,13] in partially crystallized packings. We have identified four grain-scale deformation mechanisms that recover prominent features of the diagram evolution and highlight basic grain-scale rearrangements underpinning packing crystallization and pattern selection. We note that connections between topological features of packings and their mechanical stability have recently been uncovered[54]. In this respect, we emphasize that the experimental nature of our packings ensures that the diagrams presented here actually describe the space of cavities that not only exist but are also mechanically stable. The next step would be now to explore whether other grain-scale routes towards crystallization are realizable and how mechanical constraints are embedded in the PD$_2$ representation.

The results reported here represent a basis to interpret, at the grain scale, why granular crystallization, an out-of-equilibrium phenomenon in a complex system, can be mapped onto an established framework of statistical mechanics[10,12,13]. Our findings have also practical applications in domains such as pore description in soil and geo-sciences, which are crucial for understanding natural systems mechanical stability, vibration-induced compaction and flow permeability[48,55,56].

## Methods

**Experimental protocol.** Our experimental set up extends the vertical vibrating technics used to study the compaction of random packings. The experiments are performed with beads that are covered with graphite, which was observed to reduce considerably the electrostatic repulsion between the particles. The beads are poured into two types of container geometries: cylindrical or spherical. A cylindrical container with a diameter and height of 66 mm can hold up to 300,000 beads. At this initial stage, the bead packings formed is in a random configuration with a packing density ranging from 57% to 63%.

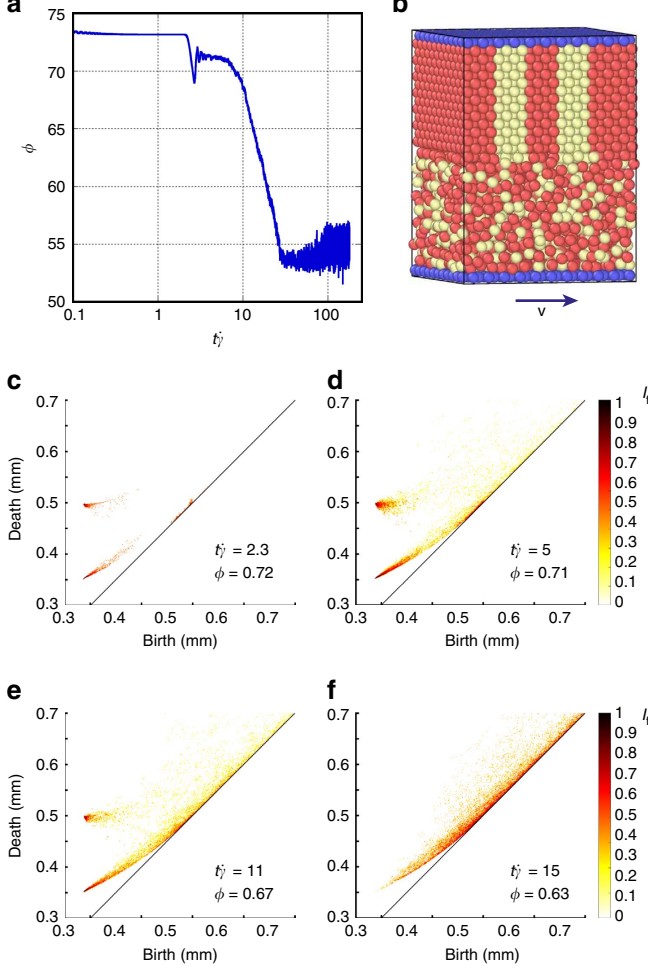

**Figure 7 | Numerical simulation of the dynamics of the order–disorder transition of a bead packing under shear.** (**a**) Packing density versus time (expressed in inverse shear rate units). (**b**) Snapshot of the numerically generated packings as it gets disordered. (**c–f**) Temporal evolution of $PD_2$ at different packing density ranging from $\phi = 0.72$ to $\phi = 0.63$. These diagrams have been computed over >6,000 beads.

To create partially crystallized packings, the whole container is placed on a shaker allowing for both vertical and horizontal vibrations. The vibrations are sinusoidal with a frequency set to $f = 50$ Hz, the vertical component of the acceleration $\gamma_v$ is set to be five times larger than the horizontal one $\gamma_h$. In these experiments, $\gamma_v$ is constant and set to $2.5g$ (where $g$ is the gravitational acceleration). The presence of 3D vibrations enhance crystallization. The container is vibrated for 20 s (1,000 periods). The resulting packings show substantial crystallization with a global packing density well beyond Bernal's limit, ranging from $\phi = 0.66$ to $\phi = 0.72$.

For $\gamma_v = 2.5g$, a collective lift off of the packing is observed in the cylindrical container. In this regime, both compaction and convection are observed[57]. The convection phenomenon plays a crucial role in the compaction/crystallization. Indeed, at lower drive, convection is absent, a very slow compaction is observed but the packing remains amorphous. In the cylindrical container, it was observed that confining the packing with a plate placed on its top enhances the crystallization process (N.B.: the plate fits perfectly in the container and can oscillate freely in the vertical direction). Measurements of the packing height during the vibration suggest that the packing remains dense (shows weak dilation) during the process; in this sense, our method is based on compaction by dense fluidization. The same observation holds for packings confined in a spherical geometry, nevertheless the convection streams always appear to be more intense in this case, which results in the formation of denser packings $\phi > 0.68$.

**Tomography and image analysis.** A typical experimental packing contains about 100,000 grains and the 3D digital image (tomogram) of the packings has a voxel size (voxel resolution) of $\approx 30$ micrometres. The beads are digitally separated by using a set of algorithms developed at ANU[58,59]. For a 1 mm diameter grain, each grain is represented by a cluster of $(4/3)\pi(33/2)^3 \approx 19,000$ voxels and each grain's

surface corresponds to a cluster of $4\pi(33/2)^2 \approx 3,400$ voxels. A grain centre is the geometric centroid of the 19,000 voxel coordinates that belong to the grain, that is, the grain centre is an average quantity computed from these large clusters of voxels that represent each grain. As a consequence of the large voxel representation of a grain's volume, the resolution on the grain centre determination is extremely high, that is, $\approx 10^{-3}$ micrometres. The precision (typical error) on the centroid determination is related to the segmentation of the voxels that cover the surface of a grain. For such a simple biphasic material, the segmentation process using our in-house software is very robust and it ensures that the precision of our measurements is comparable to our resolution within a factor of order unity[58]. To further assess the robustness of our results, we have performed topological analysis on experimental packing structures that have been postprocessed and relaxed using a discrete element method code[12,13]. $PD_2$ obtained on these numerically relaxed structures are identical to the experimental one.

As a consequence of the large voxel representation of a grain's surface, we are able to determine the average radius of a grain with a $5 \times 10^{-2}\ \mu m$ resolution. This radius has to be understood as the effective radius of an equivalent perfect sphere. By measuring the distribution of grain radii, we found that this distribution shows an average diameter of 1 mm and a width of 0.05 mm. In the main text, the width of this distribution is expressed as a 2.5% grain polydispersity.

**PH: mathematical formulation.** The starting point for computing homology is a complex, $\mathcal{C}$, essentially a collection of building blocks whose union is the shape of interest. In a simplicial complex, the building blocks are points, edges, triangles, tetrahedra and higher dimensional simplices.

A $k$-chain is a formal sum of $k$-dimensional simplices and the boundary operator is a linear map from $k$-chains to $(k-1)$-chains defined by adding up the $(k-1)$-dimensional faces of the $k$-simplices in the $k$-chain. The 'adding' is done with respect to some coefficient group; in practical applications, this is usually $\mathbb{Z}_2$, addition modulo 2. A $k$-cycle is a chain whose boundary is empty (the sum of its faces cancel out). Two $k$-cycles are said to be homologous if their difference is the boundary of a $(k+1)$-dimensional chain. The homology groups $H_k$ encode these equivalence classes of $k$-cycles.

$H_0$ represents the connected components of the simplicial complex. $H_1$ encodes equivalence classes of 1-cycles (that is, loops). Finally, $H_2$, is the equivalence classes of 2-cycles (that is, cavities).

PH extends this formalism from a single simplicial complex to a growing sequence of nested complexes called a filtration: $\{\mathcal{C}_\alpha\}_{\alpha \in \mathbb{R}}$. The complexes satisfy $\mathcal{C}_\alpha \subset \mathcal{C}_\beta$ whenever $\alpha < \beta$. The filtration parameter $\alpha$ can be a length scale or some other scalar ordering parameter. When a $k$-simplex is added to a complex in the filtration, all its faces must already be present and so the new simplex must either create a new $k$-cycle or fill in a 'hole' and make the existing $(k-1)$-cycle formed by its faces into a boundary. By tracking homologous cycles as simplices are added to the filtration, PH is able to pair the $k$-simplex that creates a $k$-cycle with the $(k+1)$-simplex that fills it in and destroys it. Each PH class therefore has two values of the filtration parameter associated with it: a birth value and a death value, as well as the actual birth and death simplices. Some cycles may be present in the final simplicial complex, these are called essential cycles and are assigned a death value of infinity. It is common practice to represent this information in a persistence diagram for each dimension of homology. $PD_k$ contains all pairs $(b, d)$, $b \leq d$, associated with PH in dimension $k$.

The simplicial complex we use for the bead packing data is built from the DT as follows. The bead packing data are specified by coordinates for the centre of each bead and its radius as extracted from micro-CT images. Recall that the definition of the DT is the union of all tetrahedra whose vertices are four data points such that their circumsphere contains no other data point. The simplicial complex contains all these tetrahedra, their triangular faces, edges and vertices. A length-scale parameter, $\alpha$, is introduced to define subsets of the DT called alpha shapes, $A(\alpha)$, that capture the topology of the union of balls of radius $\alpha$ growing around each bead centre, $X(\alpha) = \bigcup B(x, \alpha)$,[60,61]. The alpha shape contains all tetrahedra whose circumradius $\rho \leq \alpha$ and all lower dimensional simplices with circumradius less than alpha, whose circumsphere is also empty (that is, contains no other data point). Note that this empty circumsphere condition is not automatically satisfied by the lower-dimensional faces of Delaunay tetrahedra. For example, the edge opposite an obtuse angle in a triangle will have a circumsphere that contains its opposite vertex.

The filtration is the growing sequence of alpha shapes $A(\alpha)$ as $\alpha$ increases from 0 to $\infty$. Since the bead pack has a finite number of beads, the DT is finite and the topology of $A(\alpha)$ changes at a discrete set of values of $\alpha$. If we assume that the bead pack is mono-disperse with bead radius $= r$, then for $0 < \alpha < r$, $A(\alpha)$ is simply the set of data points at the bead centres. For $\alpha > r$, bead contacts are resolved and $A(\alpha)$ becomes connected, initially with many holes that are then filled in as $\alpha$ increases. For a perfectly mono-disperse bead pack with no 'rattlers', all points in $PD_0$ have birth $= 0$ and death $= r$. Points in $PD_1$ have all birth values $b \geq r$. One-cycles with $b = r$ are generated by three or four beads in contact forming a ring; those with $b > r$ are formed by triangular faces of Delaunay tetrahedra where not all four beads are in contact. $PD_2$ carries the most interesting signature of structure for the disordered and partially crystallized bead packings. Each point in $PD_2$ represents a kind of 'pore' in the interstices between the beads. The simplest and smallest pore is that formed inside four beads close packed as a regular tetrahedron. This pore is born when $\alpha$ reaches the circumradius of an equilateral triangle and dies when

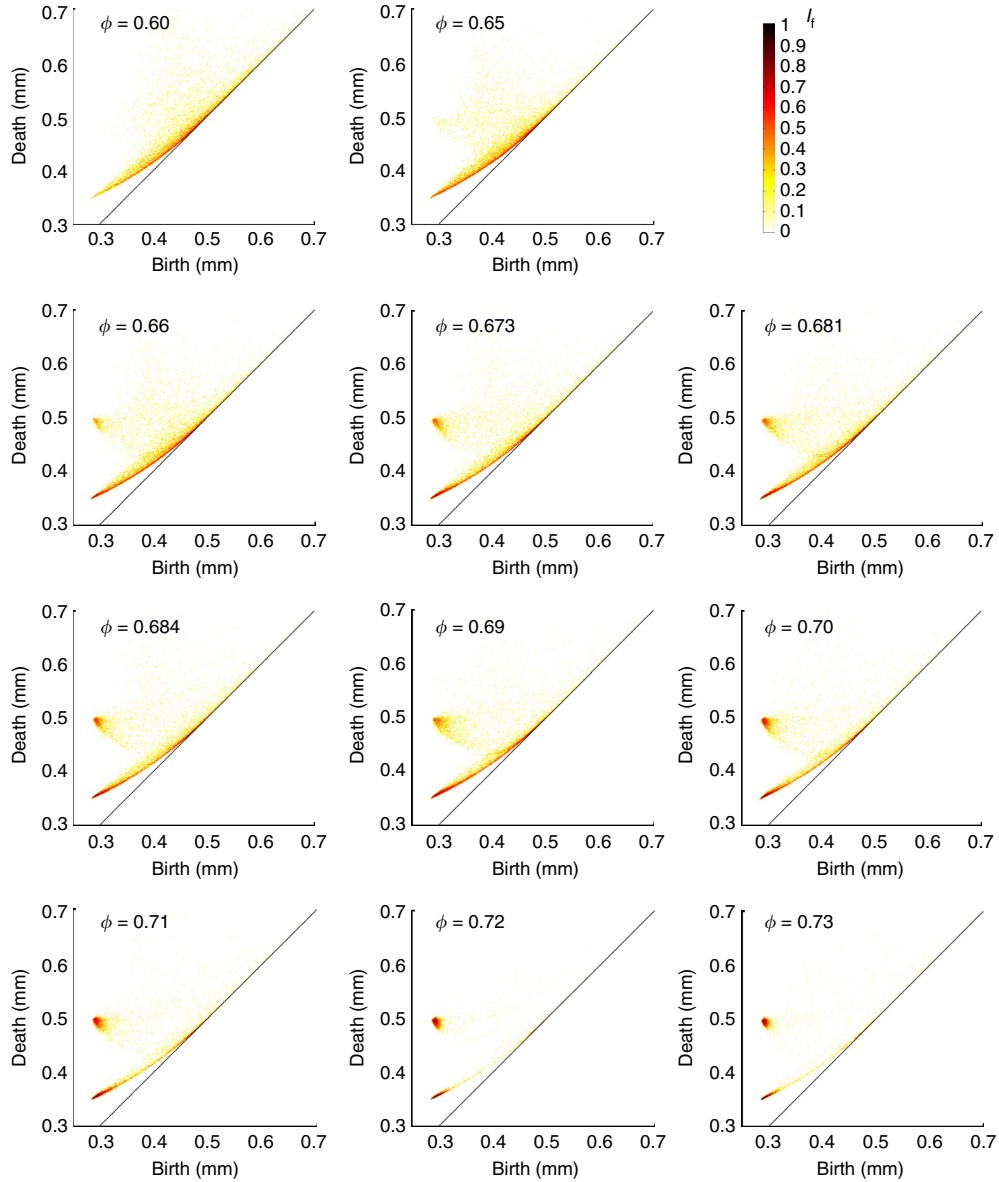

**Figure 8 | Evolution of persistence diagram as a function of packing density $\phi$.** Persistence diagrams PD$_2$ of sphere packings (subsets of 4,000 beads) over the density range $\phi = 0.60 - 0.73$.

$\alpha$ is the circumradius of the regular tetrahedron, that is, $(b, d) = (2r\sqrt{\frac{1}{3}}, 2r\sqrt{\frac{3}{8}})$. Another simple pore is that formed by six beads closely packed as a regular octahedron, with $(b, d) = (2r\sqrt{\frac{1}{3}}, r\sqrt{2})$. Spheres in a close packing with the maximal volume fraction of 0.74 have tetrahedral and octahedral pores only.

It is notable that a Delaunay tetrahedron generates a point in the PD$_2$ plot only when its circumcentre lies inside its four triangular faces. Such a tetrahedron is called 'well-centred'. The birth value is the largest circumradius of the four faces and the death value is the circumradius of the tetrahedron. Circumradii of the triangular faces, $\rho_i$, and of the tetrahedron, $R$, are related by $\rho_i = R\sin(\gamma_i)$ where $\gamma_i$ is the angle between the outward-pointing normal to face $i$ and a radial line drawn from the tetrahedron circumcentre to one of the vertices on face $i$. The angles $\gamma_i$ can take any value in $(0, \pi)$ so $\rho_i \in (0, R)$, with $\rho_i = R$ when $\gamma_i = \pi/2$ (and so have birth = death in PD$_2$). For a tetrahedron to be well-centred, however, we must have $\gamma_i < \pi/2$ for all four faces. But the largest angle in any given tetrahedron must be greater than that for a regular tetrahedron, so $\gamma_{max} \geq \arccos(\frac{1}{3})$. This means the circumradii of the faces of a well-centred tetrahedron satisfy $\rho_i \in (R\sin(\gamma_{max}), R\sin(\pi/2))$, that is, $\rho_i/R \in (\sqrt{8/3}, 1)$. This is quite a narrow strip in the persistence diagram PD$_2$ ($\sqrt{8/3} = 0.9428$), and implies that any point $(b, d)$ in PD$_2$ with $d > 3b/\sqrt{8}$ must be due to a 'pore' built from two or more Delaunay tetrahedra (that is, five or more beads in the bead-pack data).

Our bead packs are weakly polydisperse, and the variation in bead radii is compensated for by using the weighted DT (also known as a regular triangulation) based on the power distance to a ball $B(z, r)$: $d_B(x) = d(x, z)^2 - r^2$. The definition of alpha shapes carries over naturally to this setting[61].

**Analytical expressions for tetrahedral deformation.** Suppose the beads have radius $r$, then four beads can closely pack with centres at the vertices of a regular tetrahedron with edge length $2r$. As already noted, this configuration has $(b, d) = (2r/\sqrt{3}, r\sqrt{3/2})$. The deformation of the tetrahedron can be modelled by two faces remaining as equilateral triangles and one edge opening up. This can be parametrized by the dihedral angle $\theta$ opposite the opening edge and between the two equilateral faces. The birth value for this deformed tetrahedron is the circumradius of a triangle with edge lengths $2r, 2r$ and $r\sqrt{6 - 6\cos\theta}$. Using a standard formula, this gives us $b = 4r/\sqrt{10 + 6\cos\theta}$. The death value is the circumradius of the tetrahedron, so using another standard formula, with some algebra we can calculate this as $d = r\sqrt{5 - 2\cos\theta - 3\cos^2\theta}/(\sqrt{3}\sin\theta)$.

The second tetrahedral deformation is a symmetric lengthening of two opposite edges in the tetrahedron while the other four edges maintain the fixed length = $2r$. The vertices of the deformed tetrahedron may be parametrized as $(0, \pm r\sqrt{2}\cos\phi, r\sin\phi)$ and $(\pm r\sqrt{2}\cos\phi, 0, -r\sin\phi)$. The birth value is the circumradius of a triangle with edges $2r, 2r$ and $2r\sqrt{2}\cos\phi$, so $b = r\sqrt{2}/\sqrt{2 - \cos^2\phi}$. The death value is the circumradius of the tetrahedron, so $d = r\sqrt{1 + \cos^2\phi}$. These expressions can be combined to find that $d = r\sqrt{3 - 2r^2/b^2}$, which gives us an analytic expression for the second deformation pathway.

**Analytical expressions for octahedral deformation.** The first octahedral deformation is simply lengthening two opposite edges of the octahedron (symmetrically), while keeping all other edges equal to $2r$. This deformation is

parametrized by the dihedral angle $\theta$ between the faces that remain equilateral triangles. The birth value for this deformed octahedral pore is the circumradius of a triangle with edges $2r, 2r$ and $2r\sqrt{3}\cos\frac{\theta}{2}$. The standard formula evaluates to $b = 2r/\sqrt{4 - 3\cos^2\frac{\theta}{2}}$. The death value is the circumradius of the tetrahedron with four edges of length $2r$, one edge of length $2r\sqrt{3}\cos\frac{\theta}{2}$ and the opposite edge of length $2r\sqrt{3}\sin\frac{\theta}{2}$. This evaluates to:

$$d = \frac{r\sqrt{23 - 14\cos(2\theta) - 9\cos^2(2\theta)}}{4\sin\theta} \qquad (1)$$

The dihedral angle $\theta$ starts at $\left(\pi - \arccos\frac{1}{3}\right)$ (for a regular octahedron) and decreases to $\arccos\frac{1}{3}$.

The second octahedral deformation is similar to the first tetrahedral deformation (D1), where only one of the beads roll along the saddle. We can derive the following birth and death expressions: $b = r\left(4 - 2\sqrt{3}\cos\theta\right)/\sqrt{7 - 4\sqrt{3}\cos\theta}$ and $d = \frac{r}{\sqrt{3}\sin\theta}\sqrt{4 - 2\sqrt{3}\cos\theta + 3\sin^2\theta}$.

**Evolution of the persistence diagram PD$_2$ versus $\phi$.** Figure 8 shows a detailed evolution of persistence diagram PD$_2$ versus $\phi$ with emphasis over the partially crystallized density range $\phi = 0.60 - 0.73$. The figures show that as the packing density increases, the points in PD$_2$ migrate to two hot spots assosiated with the location of the tetrahedral and octahedral cavities.

**Data availability.** The data that support the findings of this study are available from the corresponding authors upon reasonable request.

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

## Acknowledgements

N.F. would like to acknowledge the support by the Australian Research Council's Discovery Early Career Research Award (DE160100742). V.R. would like to acknowledge the support by the Australian Research Council's Future Fellowship FT140100604. Y.H. acknowledges the support by JST CREST Mathematics (15656429) and JST Materials research by Information Integration Initiative. H.T. would like to acknowledge the support by JSPS KAKENHI Grant Number 16J03138. We are grateful to M. Hanifpour, S. Hyde and K. Mischaikow for fruitful discussions. M.S. thanks T. Senden for his support. We thank A. Limaye, the creator of Drishti, for his help with the 3D visualizations of the packings.

## Author contributions

M.S., N.F., V.R. and Y.H. designed research. M.S. and N.F. performed the experiments and the image analysis. H.T., V.R. and Y.H. performed the topological analysis. M.S., N.F. and V.R. wrote the paper. All authors discussed and edited the manuscript.

## Additional information

**Competing interests:** The authors declare no competing financial interests.

**Publisher's note**: 

