## [Peer Review File · Nature Communications]

Reviewers' Comments:

Reviewer #1 (Remarks to the Author):

I have reviewed manuscript "Pore deformation mechanism and configuration landscape of granular crystallization" by M. Saadatfar et. al. The manuscript describes experimental and numerical results related with the different stages displayed by a collection of spheres when the systems evolve to a crystalized state.

I found the manuscript interesting with suggestive results about the existence of dominant topological structures that determine the evolution of a grains ensemble to its final quasi-ordered state when the system is compacted. The combined use of X-Ray images and topological mathematical tools provide a useful combination in order to characterize the different motif or structures involved in the evolution of the ensemble to an ordered or crystalized final configuration.

Certainly, I can consider the publication of this manuscript but in my opinion a few important methodological and fundamental points must be clarified.

1) The robustness of the results strongly depends on the quality of the analysis of the grains spatial distribution. Indeed, the author's claims that a particle centroid can be determined within a molecular scale resolution. I've trying to follow the reference used to justify this affirmation but I cannot found any detailed analysis that justifies those results nor quantified the typical error of this value.

2) 2) What exactly means "polydispersity=0.025mm"? If I understood well they assume that all the particles are perfect spheres and only differences in its diameters are possible. It's that true? It's really astonishing for me that the author's claims that the resolution in the beads diameter is better than the typical surface roughness of standard acrylic bead. Therefore, the meaning of the term polydispersity must be clarified.

3) What happens if Euclidean distance is used instead of "power distance"? The uses of a "power distance" to characterize the packing are really poorly justified in the text. Moreover, it is not easy to follow the applicability of this magnitude from the analyses introduced in reference [70].

4) Although the idea of "crystallization" can result intuitively accessible, the concept must be fully clarified in order to justify the importance of the result. Indeed, the authors assert that the crystallization is irregular due to the polydispersity (pag. 5). It's that true? It is not conceivable perhaps, a monodisperse system with defects, like grain boundaries?

5) Regarding the PD₂ analysis, the histograms seems to be relevant to display the relative importance of different topological motif in the evolution of the packing fraction (fig. 4). Nevertheless, the authors seem to confuse the topological characterization of the different equilibrium states with the structural evolution of the grains ensemble. I agree with them that the scenery introduced in the supplementary material is feasible if quasi-static evolution of isolated group of grains is considered, but this is not the case. Although is not included in the manuscript, I assume that the protocol to access to the different packing fraction is similar to the introduced in Ref [12]: 20 sec of 2.5 g tappings. Therefore, the spatial correlation between different grains position must be completely lost. This fact is in complete contradiction with the arguments introduced at the end of the manuscript related with the role of the "Formation scenarios" described at the end of page 7. In my opinion it is completely impossible to assume that the number of "internal contacts" of any motif remains unperturbed beyond a couple of taps.

In summary, I found the application of Persistent Homology ideas to this experimental situation really valuable and interesting. Nevertheless, the theoretical interpretation developed by the authors could induce important misconceptions if the "pore deformation mechanism" introduced even in the title of the manuscript is not fully clarified.

Reviewer #2 (Remarks to the Author):

The nature of the jammed state at random close packing has been debated over long time. Here, the authors approach this problem from an experimental point of view. Different thermodynamic scenarios have been proposed theoretically and in numerical simulations to understand how a granular system transitions from a random state of packing to an order state. If understood, this problem will have large implications to different jammed systems, from granulars to glasses. Thus, the present experimental study represents an important advance in the understanding of the jammed state of matter.

The authors measure the phase space of the possible grain configurations during the crystallization of real packings that of course include polydisperse and frictional forces.

The main innovation of this study is the use of PH, which offers better capacities than the usually employed Voronoi tessellation to describe the order-disorder transition. This new technique uncovers the tetrahedral and octahedra patterns driving the crystallization

transition.

I think that this is an important contribution to the topic and should be published in NCOMM. Some suggestions for improvement follows:

1. The authors claim to resolve the particle centers at 10^{-3} microns. This is surprising. Rather than giving a reference, I am sure that the readers would like to see in the paper an explanation on how this surprising resolution of 1nm is achieved. If this resolution is true, can they extend their measurement to colloidal particles of the order of micron size? I checked ref 12 at least and I did not see an explanation of the origin of this high resolution.
2. Since the H2 group is central to the paper, please explain it better in the paper so that the reader does not need to go to the Supplementary Material for details.
3. It seems that PD2 captures the topology of the network, but how does it take into account the jamming condition between two grains that is important to characterize the jammed state?
4. It seems that the sentence: ``Moreover the existence domain of amorphous packings is still discussed:' is not well constructed grammatically.
5. Even if the authors explain well PH and PD2, it would be nice to add some wording to explain the method in a nutshell: what is the main advantage that this analysis brings to the problem?, specially in comparison to previous Voronoi tessellation approaches.
6. For instance, can they obtain the number of tetrahedra in Fig 4b with any other method, or they need to use the present topological machinery.
7. The videos do not play in Safari.

Reviewer #3 (Remarks to the Author):

This paper uses Persistent Homology to study order-disorder transition in granular packings. By focusing on tetrahedral and octahedral pores in the system, four types of mechanisms have been proposed for the evolution from the disordered pore structures to the crystalline ones. And it is claimed that one mechanism could explain why fcc will dominate in these partially crystallized packings. Although some of the results are interesting, I do feel the results are not significant enough for Nature Communications. Also, I don't get the impression that Persistent Homology is that useful for this particular system. Therefore, I would not recommend publication. Some of the concerns I had are listed in the following.

1) My biggest concern with the paper is that the whole study assumes a stroboscopic view of the order-disorder transition process. Since a large packing can have many local structures which are statistically representative of the evolution structures from the amorphous to the crystalline phase, it is then supposed that this can substitute a real dynamic study of the crystallization process. However, a first-order crystallization process is by nature a dynamic one, i.e., it involves complex nucleation, growth processes, etc. Understanding a dynamic process with a simple sampling is naturally insufficient. This makes the four mechanisms proposed more theoretical than physical. This work therefore cannot also explain where the crystallites emerge and how they grow since all these information cannot be obtained in a simple stroboscopic view, e.g., the evolution of both amorphous and crystalline structures should be heavily influenced by their local environments, therefore, if there is already one small crystal present, it will be much easier for the neighboring ones to grow upon it. The authors essentially adopted a mean-field picture by ignoring everything beyond the first shell and a local amorphous structure will simply evolve on its own. This is an oversimplification.

2) There are many different ways to characterize the local structures of amorphous packings. Like, local volume fraction, coordination number, polytetrahedra, Voronoi, Delaunay, bond orientational orders, etc. Most of these existing methods are intended to be sensitive to certain structural motifs, like icosahedra, hcp or fcc, by designating them with different numerical values. In the current study, the authors used Persistent Homology. However, my feeling is that we can probably construct a very similar evolution diagram as Fig. 3 of the current manuscript based on those existing and well-tested parameters as listed above. For example, assume a q4-q6 bond orientational order pair, or q6-w6 pair, we can probably obtain a very similar diagram. And similar structural arguments like the four mechanisms could again be used to explain the evolution. Then I didn't see how the introduction of new metrics like Persistent Homology will add to our existing knowledge. I acknowledge that studying pore structures and their connectivity are well suited to the transport properties, like thermal conductivity and permeability, of granular packings owing to their obvious relevance. However, I am not convinced of their usefulness in the current case.

3) One of the big claims of the paper is that the reason why fcc is dominant is because a pair of edge-connected tetrahedra can naturally evolve into an octahedra, it would be nice that this can be experimentally proven. However, in the current version, it looks like it is still a speculation.

Minor issue:

1) The experiment has a spatial resolution of 50 microns, and the authors claim the centers can be determined up to 10^{-3} μm , is this really true or even necessary? Since I can hardly imagine a 1mm granular particle could be perfect on 1nm scale on the surface, so any imperfection like a dent on the surface will make this resolution of the particle center quite irrelevant.

2) The whole second paragraph is related to the validity of Edwards ensemble and how friction will modify the mechanical stable structures and even the definition of the random loose packing. I see almost no relevance of this whole paragraph to the rest of the text other than the authors want to use a thermodynamic framework. It can be seriously shortened.

3) Fig.1 is almost an exact copy of a 2005 pre paper (pre 71, 061302 2005) and a 2014 prl paper (prl 113, 148001 2014) by some of the same authors, the one from pre being a mirror image and the one from prl rotated 45 degrees without even changing colors. This is kind of sloppy.

4) The Cartesian axes in all figures don't satisfy the right-hand rule.

5) In Fig. 3 and 5, the authors claim that the boundary processes correspond to the most frequent densification routes, I do see its relevance for D1, not so clearly for D2, D3 and D4 since they span a large uniform cone, so the natural question would be whether the subsequent analyses based on only these four processes can encompass all scenarios? Especially, the D3 process has been claimed as the main reason of edge-connected tetrahedra forming octahedra.

Response to Reviewers

Reviewers comments:

Reviewer #1 (Remarks to the Author):

I have reviewed manuscript "Pore deformation mechanism and configuration landscape of granular crystallization" by M. Saadafar et. al. The manuscript describes experimental and numerical results related with the different stages displayed by a collection of spheres when the systems evolve to a crystalized state.

I found the manuscript interesting with suggestive results about the existence of dominant topological structures that determine the evolution of a grains ensemble to its final quasi-ordered state when the system is compacted. The combined use of X-Ray images and topological mathematical tools provide a useful combination in order to characterize the different motif or structures involved in the evolution of the ensemble to an ordered or crystalized final configuration.

Certainly, I can consider the publication of this manuscript but in my opinion a few important methodological and fundamental points must be clarified.

A: We thank the referee for the positive and constructive review.

1) The robustness of the results strongly depends on the quality of the analysis of the grains spatial distribution. Indeed, the author's claims that a particle centroid can be determined within a molecular scale resolution. I've trying to follow the reference used to justify this affirmation but I cannot found any detailed analysis that justifies those results nor quantified the typical error of this value.

A: We thank the Referee for raising this important question. This concern is shared by all three Referees, so we now provide a new section in the Methods (Tomography and Image Analysis) to clarify how we experimentally determine each grain's centroid.

On a technical note, a typical experimental packing contains about 200,000 grains and the 3D digital image (tomogram) of the packing has a voxel resolution of 30 microns. The beads are digitally separated using a set of algorithms developed at ANU [61]. For a 1mm diameter grain, each grain's bulk is represented by a cluster of $(4/3)\pi(33/2)^3$ voxels and each grain's surface is comprised of $4\pi(33/2)^2 \sim 3,400$

voxels. The grain centres are the geometric centroid of the 19,000 voxel coordinates that belong to each grain. As a consequence of the large voxel representation of a grain's volume (~19,000 voxels), the resolution on the grain centres determination is extremely high $\sim 10^{-3}$ micron.

The precision (typical error) on the centroid determination is related to the segmentation of the voxels that compose the surface of a grain. For such a simple biphasic material, the segmentation process developed at ANU actually ensures that the precision of our measurements is comparable to our resolution within a factor unity. This remarkable feature comes from years of experience in segmenting 3D images of complex rocks Ref [58].

To further assess the robustness of our results, we have performed topological analysis on experimental packing structures that have been post-processed and "relaxed" using a discrete element method code. PDs obtained on these numerically relaxed structures are identical to the experimental ones.

2) What exactly means "polydispersity=0.025mm"? If I understood well they assume that all the particles are perfect spheres and only differences in its diameters are possible. It's that true? It's really astonishing for me that the author's claims that the resolution in the beads diameter is better than the typical surface roughness of standard acrylic bead. Therefore, the meaning of the term polydispersity must be clarified.

A: We thank the Referee for this important question on size polydispersity in our experimental packings. As a consequence of the large voxel representation of a grain's surface, we are able to determine the average radius of a grain with a 5×10^{-2} μm resolution, to be understood as the equivalent radius of a perfect sphere. By measuring the distribution of grain radii, we found that grains have an average diameter of 1mm +/- 0.025mm.

This, however, does not mean that we are able to resolve features on the surface of grains that are smaller than the voxel resolution of our 3D images (30 microns). The surface roughness can only be mapped within the voxel resolution of our images, which is ~30 microns.

We now clarify in the Methods section that the 2.5% polydispersity characterizes the width of the distribution of grains' radius, defined as an equivalent radius of a perfect sphere.

3) What happens if Euclidean distance is used instead of "power distance"? The uses of a "power distance" to characterize the packing

are really poorly justified in the text. Moreover, it is not easy to follow the applicability of this magnitude from the analyses introduced in reference [70].

A: Using the Euclidean distance does not qualitatively change the appearance of the diagrams. Since the spheres are fairly mono-disperse, the difference of derived PDs between these two distances is just a scaling factor.

The reason why the power distance was chosen is two-folds:

- First one is a technical reason: In our PH computations, we construct a weighted Voronoi decomposition and for that we need the square distances. For weighted alpha filtration (which are used in this study), the power distance formulation is quite natural since it allows to have consistent Voronoi decompositions over the filtration parameter.
- Even though grain's polydispersity is small, it remains finite. Therefore, the most faithful topological representation of our packings is based on a weighted Voronoi decomposition and the associated power distance.

We have now added new comments in the text to justify this choice (page 4, column2).

4) Although the idea of "crystallization" can result intuitively accessible, the concept must be fully clarified in order to justify the importance of the result. Indeed, the authors assert that the crystallization is irregular due to the polydispersity (pag. 5). It's that true? It is not conceivable perhaps, a monodisperse system with defects, like grain boundaries?

A: We thank the Referee for this insight. It is indeed this extremely interesting point that we are currently exploring. As seen in Figure 1, extended defects, dislocations and distorted crystal domains can clearly be observed in our system.

We emphasise again that in the context of this study, "irregularity" doesn't refer to the regularity of the packing as a whole, which for both mono- and our slightly poly-disperse systems appears as regular domains with different orientations separated by irregular grain boundaries. Our analysis is done at the pore scale and possible correlations between pore irregularity needs to be addressed with other tools. We now mention this point in the manuscript in the introduction and in page6, column 1 where we discuss the possible origins for an "irregular" crystal, where crystal domains are slightly distorted or curved.

5) Regarding the PD₂ analysis, the histograms seems to be relevant to display the relative importance of different topological motif in the evolution of the packing fraction (fig. 4). Nevertheless, the authors seem to confuse the topological characterization of the different equilibrium states with the structural evolution of the grains ensemble. I agree with them that the scenery introduced in the supplementary material is feasible if quasi-static evolution of isolated group of grains is considered, but this is not the case. Although is not included in the manuscript, I assume that the protocol to access to the different packing fraction is similar to the introduced in Ref [12]: 20 sec of 2.5 g tappings. Therefore, the spatial correlation between different grains position must be completely lost. This fact is in complete contradiction with the arguments introduced at the end of the manuscript related with the role of the "Formation scenarios" described at the end of page 7. In my opinion it is completely impossible to assume that the number of "internal contacts" of any motif remains unperturbed beyond a couple of taps.

A: The referee is quite right; our previous presentation of the data might have been confusing. In this study, experimental Persistent Diagrams are indeed measured on different static equilibrium states and it is therefore a big step to recognize that PDs can actually reveal parts of the dynamics of the crystallisation. To clarify and better convey this surprising discovery, the manuscript has been substantially modified, below we list these modifications that we feel will highlight the significance of our work:

- We now make it clear for all experimental data that we are looking at topological changes versus the packing fraction at mechanical equilibrium (i.e. statistical picture of accessible grain motifs) rather than a dynamic tracking of an evolving structure.
- We now provide results for four different partially crystallised packings produced in different container geometries. In the Methods section, we also emphasise that the shear mechanism and associated "convection" streams that induce crystallisation is dependent on the container geometry. However, PD diagrams measured at a given density are similar.
- We now provide more details on our experimental protocol and emphasise that our granular crystals are formed via shearing of a packing that remains dense when subject to the vibrations.
- Given that the configuration space of jammed matter is commonly assumed to be discrete [16], we now highlight that it is quite remarkable that our proposed continuous deformation mechanisms describe the topological domains measured on static granular states at mechanical rest (equilibrium).

- Acknowledging the Referee's comment on our experimental protocol, the discussion about the mechanical coordination number (page 7) has been removed.
- The previous discussion in page 7 has been replaced by new results from numerical simulations of frictional granular matter. These numerical experiments dynamically track the "melting" of a crystal under quasi-static shear. The evolution of PDs measured during this order-disorder transition shows features similar to the one observed experimentally in mechanical equilibrium states.

These new results suggest the possibility for an "ergodic" description of order-disorder transition in our macroscopic sphere packings.

In summary, I found the application of Persistent Homology ideas to this experimental situation really valuable and interesting. Nevertheless, the theoretical interpretation developed by the authors could induce important misconceptions if the "pore deformation mechanism" introduced even in the title of the manuscript is not fully clarified.

A: We thank the referee for this positive review. To accommodate his constructive remarks and for the sake of clarity, we have decided to modify the title to: "Pore configuration landscape of granular crystallisation". In our view, this study represents a significant step towards uncovering realistic pore deformations that seem to underlie order-disorder transition of granular crystallisation. In the introduction and conclusion, we also emphasise that these results represent a possibility for an "ergodic" description of order-disorder transition in sphere packings. We hope the new version of the manuscript better convey the importance and novelty of this discovery as well as open questions surrounding it.

Reviewer #2 (Remarks to the Author):

The nature of the jammed state at random close packing has been debated over long time. Here, the authors approach this problem from an experimental point of view. Different thermodynamic scenarios have been proposed theoretically and in numerical simulations to understand how a granular system transitions from a random state of packing to an order state. If understood, this problem will have large implications to different jammed systems, from granulars to glasses. Thus, the present experimental study represents an important advance in the understanding of the jammed state of matter.

The authors measure the phase space of the possible grain configurations during the crystallization of real packings that of course include polydisperse and frictional forces.

The main innovation of this study is the use of PH, which offers better capacities than the usually employed Voronoi tessellation to describe the order-disorder transition. This new technique uncovers the tetrahedral and octahedra patterns driving the crystallization transition.

I think that this is an important contribution to the topic and should be published in NCOMM. Some suggestions for improvement follows:

A: We thank the Referee for her/his positive review and the following constructive comments.

1. The authors claim to resolve the particle centers at 10^{-3} microns. This is surprising. Rather than giving a reference, I am sure that the readers would like to see in the paper an explanation on how this surprising resolution of 1nm is achieved. If this resolution is true, can they extend their measurement to colloidal particles of the order of micron size? I checked ref 12 at least and I did not see an explanation of the origin of this high resolution.

A: We thank the Referee for pointing out this valid concern. We have now added a new section in the Methods part providing details on how we experimentally determine the grain centroids.

On a technical note, a typical experimental packing contains about 200,000 grains and the 3D digital image (tomogram) of the packing has a voxel resolution of 30 microns. The beads are digitally separated using a set of algorithms developed at ANU [61]. For a 1mm diameter grain, each grain's bulk is represented by a cluster of $(4/3)\pi(33/2)^3$

voxels and each grain's surface is comprised of $4\pi(33/2)^2$ voxels. The grain centres are the geometric centroid of the 19,000 voxel coordinates that belong to each grain. As a consequence of the large voxel representation of a grain's volume (~19,000 voxels), the resolution on the grain centres determination is extremely high $\sim 10^{-3}$ micron.

The precision (typical error) on the centroid determination is related to the segmentation of the voxels that compose the surface of a grain. For such a simple biphasic material, the segmentation process developed at ANU actually ensures that the precision of our measurements is comparable to our resolution within a factor unity. This remarkable feature comes from years of experience in segmenting 3D images of complex rocks Ref [58].

To further assess the robustness of our results, we have performed topological analysis on experimental packing structures that have been post-processed and "relaxed" using a discrete element method code. PDs obtained on these numerically relaxed structures are identical to the experimental ones.

2. Since the H2 group is central to the paper, please explain it better in the paper so that the reader does not need to go to the Supplementary Material for details.

A: We now provide additional details on the H2 group in the main text on page 3, column 2.

3. It seems that PD2 captures the topology of the network, but how does it take into account the jamming condition between two grains that is important to characterize the jammed state?

A: A similar concern has been raised by the other two Referees. To accommodate the comments of all the Referees, we have decided to refocus the paper on the fact that PD2 captures some interesting features of the crystallisation dynamics despite being computed on static equilibrium states. Therefore, the discussion about the mechanical coordination number (page 7) has been removed and replaced by numerical experiments exploring the quasi-static "melting" a crystal under shear.

Our opinion is that PD2 provides a geometrical description with embedded mechanical constraints. We are currently exploring this question further via the new results from numerical simulations.

4. It seems that the sentence: ``Moreover the existence domain of amorphous packings is still discussed:' is not well constructed grammatically.

A: Following Referee 3's advice this paragraph has been shortened and this sentence has been removed.

5. Even if the authors explain well PH and PD2, it would be nice to add some wording to explain the method in a nutshell: what is the main advantage that this analysis brings to the problem?, specially in comparison to previous Voronoi tessellation approaches.

A: One example of the advantage of using PH is the unambiguous identification of octahedral cavities, previously only accessible indirectly through counts of "quart-octahedral" tetrahedra, which may not have all belonged to octahedra.

The basis for computing PH is the alpha shape (Delaunay tessellation with tetrahedra ordered by their circumradii) and therefore it integrates the geometry as well as local topology of grain structures. In short PH naturally incorporates the relevant local correlations in bead positions to reveal short and medium-range order, and even some global structures (the percolating length scales).

We now provide additional details on our topological method in the main text on page 4, column 2.

6. For instance, can they obtain the number of tetrahedra in Fig 4b with any other method, or they need to use the present topological machinery.

A: Of course, the definition of tetrahedra, or to be precise, quasi-regular tetrahedra is possible via Delaunay triangulation but it remains a difficult topic. Many metrics have been proposed, and the work of Hales is now the reference. In our study, the comparison between results from PH and the standard methods based on Delaunay tessellation give different results, see comments and Ref [11] on page 6, column 2. One of our future research directions is to investigate why such differences exist. Again, PH is also able to identify unambiguously the local octahedral cavities which is a clear asset compared with the methods based on Delaunay partition only.

7. The videos do not play in Safari.

A: The videos are in AVI format. We have re-created the videos and we confirm that they play in QuickTime and VLC.

Reviewer #3 (Remarks to the Author):

This paper uses Persistent Homology to study order-disorder transition in granular packings. By focusing on tetrahedral and octahedral pores in the system, four types of mechanisms have been proposed for the evolution from the disordered pore structures to the crystalline ones. And it is claimed that one mechanism could explain why fcc will dominate in these partially crystalized packings. Although some of the results are interesting, I do feel the results are not significant enough for Nature Communications. Also, I don't get the impression that Persistent Homology is that useful for this particular system. Therefore, I would not recommend publication. Some of the concerns I had are listed in the following.

A: We thank the referee for her/his critical review. The Referee questions the novelty and relevance of our study to the topic of crystallisation in a frictional sphere packings and its significance to answer broad and challenging questions related to out of equilibrium physics in general.

We believe that our results are significant due to the following reasons:

- Creating the basis for a statistical description of order-disorder transition in a macroscopic dissipative material is a fundamental challenge of physics; moreover, although exciting analogies between order-disorder transition in granular layers and thermodynamic transition have been recently uncovered, an analogy is not an equivalence. As such, understanding of the structure of partially crystallised granular media is a central and open problem for a broad community and for the granular community in particular.
- The goal of statistical mechanics in general is to prove that average observables can describe a complex dynamical process. Our new results reveal a possible ergodicity in the crystal growth in frictional sphere packings, which would strongly support a thermodynamic-like description of this out of equilibrium process. Indeed, our paper is paving the way for such a description via "average", statistically representative deformation mechanisms. The deformation of tetrahedral and octahedral arrangements of grains shown in this work are not theoretical, they are realistic and have been deduced directly from actual crystallisation processes in dissipative granular media.

The Reviewer questions whether Persistent Homology is a suitable method to study granular materials. On the contrary, Persistent Homology is actually essential and even unique. It provides a much broader and richer picture of the configurational landscape than any other methods known.

This is essentially at the heart of this study: a statistical description might be relevant to describe the complex dynamic of crystallisation in sphere packings and Persistent Homology offers an exciting and unique mapping of the associated configuration landscape. To better convey this message and accommodate the referee's concerns, we have made multiple modifications to the manuscript, analyse new experiments, and perform new numerical simulations where we can dynamically measure PD2 during an order-disorder transition. All results are consistent with the original message of our study. Below we address the Reviewer's major and minor comments.

1) My biggest concern with the paper is that the whole study assumes a stroboscopic view of the order-disorder transition process. Since a large packing can have many local structures which are statistically representative of the evolution structures from the amorphous to the crystalline phase, it is then supposed that this can substitute a real dynamic study of the crystallization process. However, a first-order crystallization process is by nature a dynamic one, i.e., it involves complex nucleation, growth processes, etc. Understanding a dynamic process with a simple sampling is naturally insufficient. This makes the four mechanisms proposed more theoretical than physical. This work therefore cannot also explain where the crystallites emerge and how they grow since all these information cannot be obtained in a simple stroboscopic view, e.g., the evolution of both amorphous and crystalline structures should be heavily influenced by their local environments, therefore, if there is already one small crystal present, it will be much easier for the neighboring ones to grow upon it. The authors essentially adopted a mean-field picture by ignoring everything beyond the first shell and a local amorphous structure will simply evolve on its own. This is an oversimplification.

A: We understand the Referee's concerns. In the previous version of the paper, it was difficult to judge whether the PD diagrams could actually capture some of the dynamics of crystal growth and to grasp what the grain-scale deformation mechanisms mean. Most of the confusion stems from the discussion on the mechanical features of partially crystallised packings, we have therefore decided to remove it and replace it by completely new results obtained from numerical simulations.

To be more specific the reviewer raises two important questions here:

1. Crystallisation is by nature a dynamic process that cannot be understood by a simple sampling, therefore our results are more theoretical than experimental.
2. Our work cannot describe where the first stage of crystallisation occurs, namely the nucleation of a crystallite nor are we able to follow its growth; therefore, it would not be possible to describe some relevant process of crystal growth.

Below we respond to each of the above points.

1:

1a. First to accommodate the referee's concerns about a single sampling, we have now performed three new experiments in different container geometry and with different beads diameter. The partially crystallised packings studied account for more than 4×10^5

beads. As can be seen in the new figure 3, the PD2 are similar for all these packings.

1b. We now make it clear in all the experimental part that we are looking at topological changes versus the packing fraction at static equilibrium rather than a dynamic tracking of an evolving structure.

1c. We now provide new results from numerical simulations in which we actually dynamically track the “melting” of a granular crystal under a quasi-static shear. During this order-disorder transition, the evolution of PD2 is similar to the one observed experimentally in a mechanical equilibrium state.

2:

The referee is right; this study is not about grain tracking during granular crystallisation. We acknowledge that some of the wording used in the previous version of the paper and our former discussion on page 7 might have been confusing, this has been corrected throughout the paper. However, we feel that this fair concern raised by the referee lead her/him to largely underestimate the significance of the results. Although we are not following the growth of a crystallite, i.e. a crystalline “grain”, we do describe crystal growth at the grain-scale. We believe it is misleading and wrong to convey the idea that order-disorder transition (especially a process that presents analogies with a first-order transition [12,13]) depends dramatically on where the formation of crystallite occurs. In this respect, our study does not deal with the question of *where* but with the question of *how* a large crystal forms?

Our experimental data do capture essential information on the different stages of crystal formation as enumerated by the reviewer. This claim is now further supported by new results from numerical simulations that provide the dynamical topological signature in PD2 of the “melting” of a crystal. Moreover, these new results support a possible “ergodicity” of the crystallisation process in frictional sphere packing.

All the formation scenarios uncovered by our PD analysis are clearly relevant to how a crystallite might form. While crystallization proceeds, crystallites grow in size or merge and our results can also be understood as a grain-scale description of these growth mechanisms of large crystalline clusters. Given their experimental nature and being based on large polycrystalline samples, these diagrams also capture the interaction of multiple crystalline domains: i.e. the influence of the local environment.

Again we acknowledge the confusion that has stemmed from our previous discussion and which has now been removed. We now fully address in the manuscript the Referee’s concerns, by analysing new

experimental results (new figure 3), new numerical results described in Figure 7 and by substantially changing the final discussion according to these new results. Our study should thus be considered as one of the first experimental investigation of the configuration space during crystallisation in 3D frictional sphere packings.

2) There are many different ways to characterize the local structures of amorphous packings. Like, local volume fraction, coordination number, polytetrahedra, Voronoi, Delaunay, bond orientational orders, etc. Most of these existing methods are intended to be sensitive to certain structural motifs, like icosahedra, hcp or fcc, by designating them with different numerical values. In the current study, the authors used Persistent Homology. However, my feeling is that we can probably construct a very similar evolution diagram as Fig. 3 of the current manuscript based on those existing and well-tested parameters as listed above. For example, assume a q4-q6 bond orientational order pair, or q6-w6 pair, we can probably obtain a very similar diagram. And similar structural arguments like the four mechanisms could again be used to explain the evolution. Then I didn't see how the introduction of new metrics like Persistent Homology will add to our existing knowledge. I acknowledge that studying pore structures and their connectivity are well suited to the transport properties, like thermal conductivity and permeability, of granular packings owing to their obvious relevance. However, I am not convinced of their usefulness in the current case.

A: The referee expresses a “negative feeling” on our approach, a feeling based on the idea that more common geometric approaches such as bond order parameters could provide the same results. However, by doing so (s)he also acknowledges that no one has actually been able to perform a study like ours. There is a good reason for that and it is related to the power, versatility and novelty of Persistent Homology.

It is true that some of the information presented here can be recovered through some other known methods such as q4-q6, we argue that PH provides a much broader and richer picture of the configurational landscape than any other of those methods. Below we detail some of the shortcomings of the q4-q6 Bond Order techniques and the reasons why PH is superior.

There are two main issues with the Bond Order characterisation tools:

1. The “ambiguity of the neighbourhood definition”: The choice of a set of nearest neighbours (at the heart of bond orientation analysis) is not unique and quite arbitrary. For example:

- Steinhardt et al. proposed to use “some suitable set” of bonds for the computation of q_l ; they used a definition based on a cut-off radius of $1.2d$, where “ d ” is grain diameter. Neighbourhood definitions based on cut-off radii are widely used, e.g., with cut-off radii $1.2d$ and $1.4d$ or with the value of the cut-off determined by the first minimum of the two-point correlation function $g(r)$.
 - Neighbours have been defined using the Delaunay graph of the particle centres.
 - A fixed number NN of neighbours is assigned to each particle ($NN=12$ in 3D).
2. Bond Order is a Short Range order parameter dealing with the nearest neighbours (first shell). However, since amorphous structures lack pre-determined structural signatures (periodicity etc.) of crystals, one needs to map and decipher the mid to long range order and the hierarchical structures. PH gives us the ability to map the hierarchy of structure at mid to long range beyond the first shell.

Finally, we strongly disagree with the Reviewer’s last comment: our new analysis clearly adds to the existing knowledge. There are only a handful of experimental references on order-disorder transition in 3D packings. We would like to re-iterate that our study uncovers new underlying geometrical and topological changes that drive crystallisation. Such detailed grain-scale description of order-disorder transition in dissipative macroscopic granular media is unknown to date despite its relevance and importance to many scientific and engineering domains. We propose four deformation scenarios that occur in real granular materials that could not have been uncovered without the aid of Persistent Homology.

3) One of the big claims of the paper is that the reason why fcc is dominant is because a pair of edge-connected tetrahedra can naturally evolve into an octahedra, it would be nice that this can be experimentally proven. However, in the current version, it looks like it is still a speculation.

A: We do acknowledge the Referee’s frustration with the fact that experimentally we are unable to dynamically track the D3 scenario as it occurs in an evolving granular packing. At present with currently available 4D imaging technologies we do not have sufficient temporal resolution to access such grain scale information. However, based on our experimental observation from static granular packings at various

packing densities, we infer the existence of D3 scenario. We cautiously present the tantalising possibility that edge-connected tetrahedra can naturally deform into an octahedron (the D3 scenario). For instance we state in the abstract that that “these mechanisms ... give clues to interpret the observed dominance of FCC motifs”.

We agree with the Referee and that is why in the revised manuscript we keep a measured tone when addressing this question. However, to date we are not aware of any constructive attempt to explain why FCC is a preferred motif in frictional macroscopic 3D packings. Our new numerical results support a possible ergodicity of the transition and they support the relevance of D3 to tackle this question. Any attempt towards this goal should thus be welcomed and to quote the Referee “*a large packing can have many local structure which are statistically representative of the evolution of structures*”. This is exactly what D3 scenario is about, a statistically representative path of evolution between HCP and FCC motifs.

Minor issue:

1) The experiment has a spatial resolution of 50 microns, and the authors claim the centers can be determined up to 10^{-3} μm , is this really true or even necessary? Since I can hardly imagine a 1mm granular particle could be perfect on 1nm scale on the surface, so any imperfection like a dent on the surface will make this resolution of the particle center quite irrelevant.

A: Reporting experimental measurements as they are is essential. We now provide details in the supplementary part on how we experimentally determine the grain centroids. To further assess the robustness of our results, we have performed topological analysis on experimental packing structures that have been post-processed and “relaxed” using a discrete element method code. PDs obtained on these numerically relaxed structures are identical to the experimental one.

On a side note, a 1nm dent on the surface will barely make a “dent” in the resolution of centre measurement and is even less likely to affect a pore formed by 1mm diameter beads.

2) The whole second paragraph is related to the validity of Edwards ensemble and how friction will modify the mechanical stable structures and even the definition of the random loose packing. I see almost no relevance of this whole paragraph to the rest of the text other than the authors want to use a thermodynamic framework. It can be seriously shortened.

A: The paragraph has now been shortened but the main message has been preserved, i.e. building a strong basis for a statistical description of order-disorder transition in a dissipative granular material. This approach heavily relies on the description of the configuration landscape of sphere packings that is the main message of the paper. In that respect, the work of Edwards and co-workers is obviously relevant and this inspiring context has to be acknowledged.

3) Fig.1 is almost an exact copy of a 2005 pre paper (pre 71, 061302 2005) and a 2014 prl paper (prl 113, 148001 2014) by some of the same authors, the one from pre being a mirror image and the one from prl rotated 45 degrees without even changing colors. This is kind of sloppy.

A: This has now been corrected. We now provide new visualisations that highlight the heterogeneous structure of our packings, where random and crystalline clusters coexist. Moreover, the new figure 1(b) offers a better illustration of the polycrystalline nature of our packings.

4) The Cartesian axes in all figures don't satisfy the right-hand rule.

A: We thank the Referee for picking this up. This has now been corrected in Figure 2 and Figure 6.

5) In Fig. 3 and 5, the authors claim that the boundary processes correspond to the most frequent densification routes, I do see its relevance for D1, not so clearly for D2, D3 and D4 since they span a large uniform cone, so the natural question would be whether the subsequent analyses based on only these four processes can encompass all scenarios? Especially, the D3 process has been claimed as the main reason of edge-connected tetrahedra forming octahedra.

A: This is an interesting question indeed. We have tried many other "natural" deformation scenarios but none of them match with the existence domain (see comments on page 8, column 2). PH is clearly a great tool to explore other routes and grain-scale scenarios to crystallisation, we have added a comment on this topic in the conclusion. Nevertheless, we'd like to emphasise that the population parameter " f " in the PDs is plotted on a loglog scale. As such, the fact that curves D3, D4 highlight contours that are clearly red while the inside of the cone is obviously yellow, implies that these curves are much more densely populated (i.e. have much higher statistical weight)

than the latter. In the manuscript we also emphasise their importance/special role as “limiting curves” / boundaries of the existence domain.

Reviewers' Comments:

Reviewer #1 (Remarks to the Author):

I have reviewed the renewed version of the manuscript NCOMMS-16-06935A by M. Saadatfar et. al.

In the new version, the author clarifies my doubts about its calculations, deepening in the meaning of its results.

Hence, I hope that this work will stimulate new microscopic approaches to the study of the particle ensembles and in consequence, I recommend its publication in Nature Communications.

Reviewer #2 (Remarks to the Author):

The authors have adequately responded to all my concerns. I believe that the introduction of a new topological parameter to characterize ordering transitions in packings is a significant result, as important as the introduction of other celebrated metrics like the q_m bond ordering parameters of Steinhart et al.

Thus, I believe that this paper should be of interest to the community at large, not only in jamming but physics in general, and therefore I recommend its publication in NComm.

Reviewer #3 (Remarks to the Author):

The authors provides a more convinving argument on the crystalization process in mechanical stable granular packings, although I feel a global growth process will need a dynamic study. But how individual event happens could indeed be quite localized and controlled by certain universal pathways. these unique pathways could be related to the saddle directions of landscape just at the crystallization point which can reveal important info on its topology, therefore the study has some value in this sense. Therefore, I would recommend publication.